# Comparative Study for Characteristics of Locomotive Syndrome in Patients with Lumbar Stenosis and Adult Spinal Deformity

**DOI:** 10.3390/jcm12134345

**Published:** 2023-06-28

**Authors:** Tetsuro Ohba, Go Goto, Kotaro Oda, Nobuki Tanaka, Hiroshi Yokomichi, Hirotaka Haro

**Affiliations:** 1Department of Orthopaedics, University of Yamanashi, Shimokato, Chuo 409-3898, Yamanashi, Japan; goto.gtg97@gmail.com (G.G.); koda@yamanashi.ac.jp (K.O.); tanakan@yamanashi.ac.jp (N.T.); haro@yamanashi.ac.jp (H.H.); 2Department of Health Sciences, University of Yamanashi, Shimokato, Chuo 409-3898, Yamanashi, Japan; hyokomichi@yamanashi.ac.jp

**Keywords:** adult spinal deformity, lumbar spinal stenosis, locomotive Syndrome

## Abstract

Introduction: The differential diagnoses of lumbar spinal stenosis (LSS) and adult spinal deformity (ASD) have been demonstrated primarily using sagittal radiographic spinopelvic parameters. However, it is more important to know the differences in the characteristic clinical symptoms to make accurate treatment decisions. Recently, the relationship between spinal disease and Locomotive Syndrome (LS) has been reported. Additionally, the Geriatric Locomotive Function Scale-25 (GLFS-25) was reported to be a useful scale to evaluate disease severity and characteristic clinical symptoms in spinal disease. Methods: Sixty-nine consecutive patients with ASD and 196 patients with LSS who underwent spinal surgery were included. Locomotive dysfunction was evaluated using the GLFS-25 questionnaire and physical performance tests including the two-step test and the stand-up test, measured preoperatively. The correlations between sagittal spinopelvic parameters of ASD and LS were examined. Results: All subjects with lumbar degenerative disease in the present study were diagnosed with LS preoperatively. The severity of LS in patients with LSS and ASD were statistically similar. GLFS-25 scores in the mobility and community domain were similarly poor in both groups. Several scores in the domestic life and self-care domains were significantly worse in the ASD group. Question 20 of the GLFS-25, related to load-bearing tasks and housework, was significantly associated with a large pelvic incidence in ASD patients. Conclusions: Lumbar degenerative disease requiring surgery severely affects the LS of older people. ASD patients had more difficulty with load-bearing tasks and housework such as cleaning the yard, carrying heavy bedding, dressing, and bathing compared to LSS patients.

## 1. Introduction

With an aging population, there is an increasing opportunity to treat musculoskeletal diseases including spinal degenerative diseases such as lumbar spinal stenosis (LSS) and adult spinal deformities (ASD). Both LSS and ASD have been known to have a great influence on health-related quality of life. These diseases are known to be correlated with pain as well as physical, mental, and internal functional disability. Generally, conservative treatments such as medication and exercise are implemented for these diseases, but if symptoms do not improve for a long period of time, the physical ability may be compromised. Therefore, surgical treatment is considered for cases that do not improve with conservative treatment. The differential diagnosis is important because surgical invasion and strategy differ greatly depending on whether surgery is performed on LSS or ASD patients. Sagittal spinopelvic alignment parameters with full-length frontal and lateral radiographs are essential for the diagnosis of these conditions, and their progression is used frequently to determine the surgical formula. In practice, the surgeon combines imaging diagnosis with clinical findings to determine the surgical procedure. Therefore, it is important to distinguish the characteristic clinical symptoms and chief complaints for each of these diseases.

In a society in which the average age is increasing, elderly people requiring care services due to declining physical ability will be also increasing. Past studies suggested that remaining physically active is associated with less morbidity and mortality [1]. Adapting to an aging society, the Japanese Orthopaedic Association (JOA) has proposed the concept of a “Locomotive Syndrome (LS)” to designate the condition of elderly patients with musculoskeletal disease who are highly likely to require nursing care [2]. There have been many reports on the relationship between spinal disease and LS. Recently, a Geriatric Locomotive Function Scale (GLFS-25) consisting of 25 simple questions have been developed for the early detection of LS and reported as a useful scale to evaluate disease severity in spinal disease [3,4].

To establish the optimal surgical strategy for treating ASD and/or LSS, we (1) compared the prevalence and severity of LS in LSS patients without sagittal imbalance and ASD patients who were treated with spinal surgery and (2) identified the symptoms characteristic of each disease using the GLFS-25.

## 2. Materials and Methods

### 2.1. Patients

After approval by our institutional review board, we conducted a retrospective observational study of a cohort of consecutive patients with a diagnosis of ASD or LSS who underwent spinal surgery. Patients with ASD were considered candidates for thoracolumbar correction if spinal fusion was an indicated treatment and if a full course of conservative care had been exhausted. The inclusion criteria were patients > 60 years old with a radiographic diagnosis of ASD defined by at least one of the following parameters: a coronal Cobb angle > 30°; a C7 sagittal vertical axis (SVA) > 50 mm between the C7 plumb line and the postero-superior edge of S1; and/or >30° pelvic tilt (PT) defined as the orientation of the pelvis with respect to the femurs and the rest of the body. A SVA of <50 mm, a lumbar lordosis (LL) of >30°, and/or a Cobb angle of <10° were defined as lumbar spinal stenosis (LSS). We included data from 69 consecutive patients with ASD and 196 consecutive patients with LSS, all of whom underwent spinal surgery for their disorder between April 2018 and March 2022. All surgeries were performed by two board-certified spinal surgeons at a single institution. Age, sex, and body mass index (BMI) were recorded (Table 1). All data pertaining to the physical performance tests and LS were collected preoperatively.

### 2.2. Radiographic Measurements

Preoperative radiographic data were acquired from full-length frontal and lateral radiographs with the patient in a freestanding posture with their fingers placed on their clavicles. All radiographic parameters were measured using a lateral view: T5–T12 thoracolumbar kyphotic curvature; T12–S1 lumbar lordosis (LL) angles; pelvic incidence (PI); pelvic tilt (PT); sacral slope (SS); sagittal vertical axis (SVA); T1–pelvic angle (TPA) defined as the angle between the line from the center of the femoral heads to the center of S1 and the line from the femoral head to the center of the T1 vertebra global tilt (GT), defined as the angle formed by the intersection of two lines, the first drawn from the center of C7 to the center of the sacral endplate and the second drawn from the center of the femoral heads to the center of the sacral endplate [5]. Kyphosis is expressed as a positive value and lordosis is expressed as a negative value. Radiographic measurements were obtained by two board-certified spinal surgeons (Authors 1 and 5) to determine interobserver error. The mean values of their measurements were applied in our analysis. The intraclass coefficient was 0.890, indicating that the inter-rater reliability was almost ideal. The two surgeons have >10 years of experience in spinal surgery and were blinded to all patient data before any measurements were performed.

### 2.3. LS Screening Instrument

The Geriatric Locomotive Function Scale-25 (GLFS-25) is a self-administered 25-item questionnaire initially developed to screen elderly adults with locomotive dysfunction [3]. It consists of questions about pain, self-care, mobility, locomotion, housework, social activity, and anxiety. Responses to individual questions are graded on a scale of 0 to 4 points classified as follows: 0 points, no impairment and not difficult to do; 1 point, mild impairment and mildly difficult to do; 2 points, moderate impairment and moderately difficult to do; 3 points, considerable impairment and considerably difficult to do; 4 points, severe impairment and extremely difficult to do. All points were added together to determine a total score. The total GLFS-25 score ranges from 0 to 100. GLFS-25 was collected preoperatively. According to a previous study, the severity of the LS that was diagnosed was based on cutoff scores for total GLFS-25, which were 7/100 for grade 1 LS (LS1), 16/100 for grade 2 LS (LS2), and 24/100 for grade 3 (LS3) [3].

### 2.4. Physical Performance Tests

Two tests, proposed by the JOA, were used for assessment of LS [6].

(1)Two-step test: a measurement of the length of two strides of the participants, providing a general assessment of their walking ability including muscular strength, balance, and flexibility of the lower extremities. It is scored by normalizing the maximal length of two steps by the height. Two-Step Test scores < 0.9, <1.1, <1.3, and ≥1.3 points correspond to LS-3, LS-2, LS-1, and non-LS [7].(2)The stand-up test: a simple method of assessing the muscular strength of the lower extremities by having participants stand-up on one or both legs once from seats of different heights (40, 30, 20, and 10 cm). The test is scored as 0–8, with the scores defined as follows: 0 (unable to stand); 1–4 (able to stand—using both legs—from 40, 30, 20, and 10 cm, respectively); 5–8 (able to stand—using one leg—from 40, 30, 20, and 10 cm, respectively). Stand-Up Test scores of 0–1, 2, 3–4, and 5–8 points are equivalent to LS-3, LS-2, LS-1, and non-LS [7].

### 2.5. Statistical Analyses

Mean ± SD values are reported for continuous variables. Number (percentage) values were used for categorical variables. We performed Student’s *t*-tests, the Mann–Whitney test, or Fisher’s exact test to compare the mean values between the two groups (pre- and post-surgical patients), assuming normal distributions for continuous variables. The relationships between spinal pelvic parameters of LSS and ASD patients and responses to the GLFS-25 questions were determined using Pearson correlation coefficients. We used Prism (version 8.0; GraphPad Software, La Jolla, CA, USA) to calculate summary statistics and perform the *t* tests. Asterisks indicate statistical significance (*p* < 0.05).

## 3. Patient Population

There were no significant differences between the LSS and ASD groups for age, gender, or body mass index (BMI) (Table 1). Among physical performance tests, there were no significant differences in the two-step test or the stand-up test between groups (Table 1).

## 4. Frequency and Severity of Locomotive Dysfunction in Patients with LSS or ASD

The prevalence of LS in both LSS and ASD patients before surgery was 100%. The frequency of LS3 was 81.1% in LSS patients and 85.5% in ASD patients. The frequency of LS2 was 12.7% in LSS patients and 6.3% in ASD patients. The frequency of LS1 was 6.1% in LSS patients and 5.8% in ASD patients. There was no significant difference of severity of LS between groups (Figure 1).

## 5. GLFS-25 Scores in Patients with LSS or ASD

The total score on the GLFS-25 was not significantly different between groups (Figure 2). Pain in the lower limbs (Q3) was significantly worse in the LSS group compared to the ASD group (Table 2). Mobility domains were equally poor in both groups. In contrast, some questions about self-care were significantly worse in the ASD group compared to the LSS group (Q8, 11, 14). ASD patients have more difficulty with load-bearing tasks and housework such as cleaning the yard, carrying heavy bedding, dressing, and bathing (Q20) compared to LSS patients (Table 2).

## 6. Correlation between GLFS-Scores and Spinopelvic Parameters in ASD

The correlation between total score of GLFS-25 and spinopelvic parameters was no significant. Only one GLFS-25 question (Q20), related to load-bearing tasks and housework, was significantly associated with spinopelvic parameters in ASD patients (a large PI, Table 3). There was no significant correlation between any of the other questions and spinopelvic parameters in LSS and ASD patients.

## 7. Discussion

Sagittal spinopelvic alignment parameters with full-length frontal and lateral radiographs are standard methods for the diagnosis of these conditions, which is used frequently to determine the surgical formula. However, extreme care should be taken not to decide on a surgical formula based on diagnosis from static imaging alone [8]. Decision-making and treatment options for LSS with sagittal imbalance are still controversial [9]. In fact, significant improvement of sagittal spinopelvic alignment has been known after simple lumbar decompression surgery for LSS [10]. Therefore, it is important to distinguish the characteristic clinical symptoms and chief complaints for each of these diseases.

Surprisingly, all subjects with the lumbar degenerative disease in this study were diagnosed with LS preoperatively. This result indicated that lumbar degenerative disease requiring surgery severely affects LS of elders. Additionally, the study showed that the severity of the LS in patients with LSS and ASD was statistically similar. According to a detailed evaluation of the questions of the GLFS-25, scores in the mobility and community domain were poor in both groups. Only lower limb pain (Q3) was significantly worse for the LSS patients. In contrast, several questions on domestic life and self-care were significantly worse in the ASD group. Typical symptoms characteristic of LSS might be neurogenic intermittent claudication, lower extremity pain, and numbness and weakness in the lower extremities [11]. In contrast, back pain and standing gait disturbance due to forward-stooped posture is one of the representative symptoms of ASD [12]. Additionally, adult spinal deformity has been reported to be correlated with many symptoms not only of musculoskeletal disorders but also possibly of disorders of visceral organs close to the spine such as gastroesophageal reflux disease and pulmonary dysfunction [13,14].

These symptoms might be determining factors indicating the need for corrective spinal fusion surgery. A recent study reported that ASD patients performed poorly on functional mobility tests including the alternate step test, six-meter walk test, sit-to-stand test and timed up and go test compared to LSS patients [8]. Consistent with this result, the the present study clarified that ASD patients have more difficulty with load-bearing tasks and housework such as cleaning the yard, carrying heavy bedding, dressing, and bathing compared to LSS patients. Although there have been previous reports focusing on activities of daily living after spinal corrective surgery [15] to our knowledge this is the first study showing that ASD patients were more impaired in some activities of daily living compared to LSS patients before surgery.

The close relationship between spinal sagittal imbalance and LS has been described [5,16]. Several spinopelvic parameters including SVA, PT, SS, and PI-LL showed an association with GLFS-25 [16,17,18]. In the present study, only one GLFS-25 question (Q20), related to load-bearing tasks and housework, (Q20) was significantly associated with any spinopelvic parameter (a large PI) in ASD patients. This result indicated pelvic malalignment might make it difficult to perform mid-back and lumbar spine-bearing tasks. We need further investigation as to why we did not find a correlation between any questionnaire other than Q20 and the spinal pelvic parameters in present study. Various types of spinal deformity, such as lumbar spine kyphosis without thoracic spine degeneration, and large rigid kyphosis of the thoracic spine, were included in the present study. Therefore, a more detailed study using a larger sample should be conducted in the future. Another hypothesis for the lack of a strong correlation between GLFS-25 scores and spinopelvic parameters in present study is that the patients in this present may have been severe symptoms who required surgical treatment. In fact, about 80% of the subjects in this study had the most severe grade 3 LS. Past studies using volunteers have reported correlations between locomotion and spinopelvic parameters and improvement of the sagittal spinopelvic alignment and balance through therapeutic exercise [5,16,19]. Early diagnosis and early intervention are important for the treatment of LS and future research is needed.

Present study has some limitations. Firstly, this is small number and retrospective observational study. Secondly, present study did not analyze the impact of surgery or conservative treatment on GLFS-25 scores and/or LS severity. Previously, we showed that despite spinal corrective surgery resulting in significantly lower frequency and severity of LS in patients with ASD, some GLFS-25 items can worsen after surgery [17]. In the future, it is necessary to study the difference in postoperative outcomes between LSS and ASD. However, the present study has clinical significance as there are some differences in the characteristics of disability in activities of daily living between LSS and ASD.

## Figures and Tables

**Figure 1 jcm-12-04345-f001:**
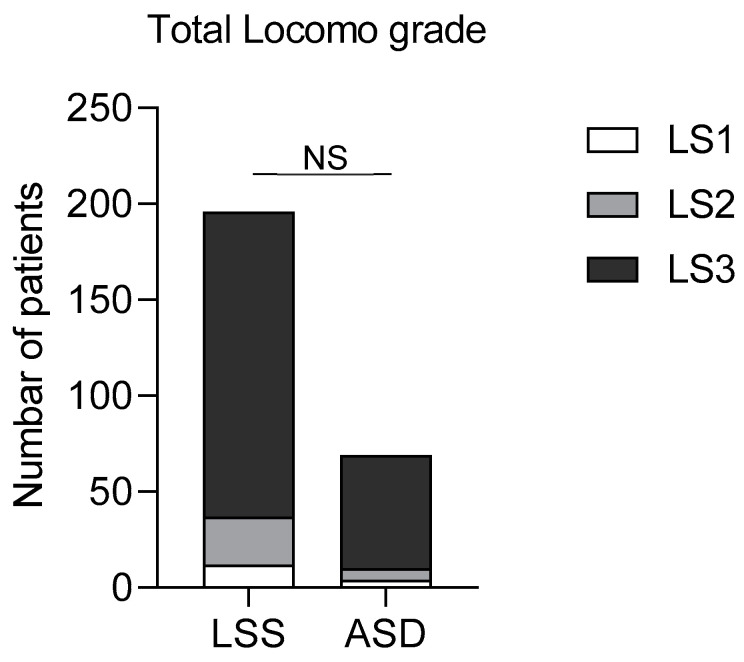
Frequency and severity of locomotive dysfunction in patients with LSS or ASD. NS; No significant; LSS: Lumbar spinal stenosis; ASD: Adult spinal deformities; LS1: Locomo grade 1; LS2: Locomo grade 2; LS3: Locomo grade 3.

**Figure 2 jcm-12-04345-f002:**
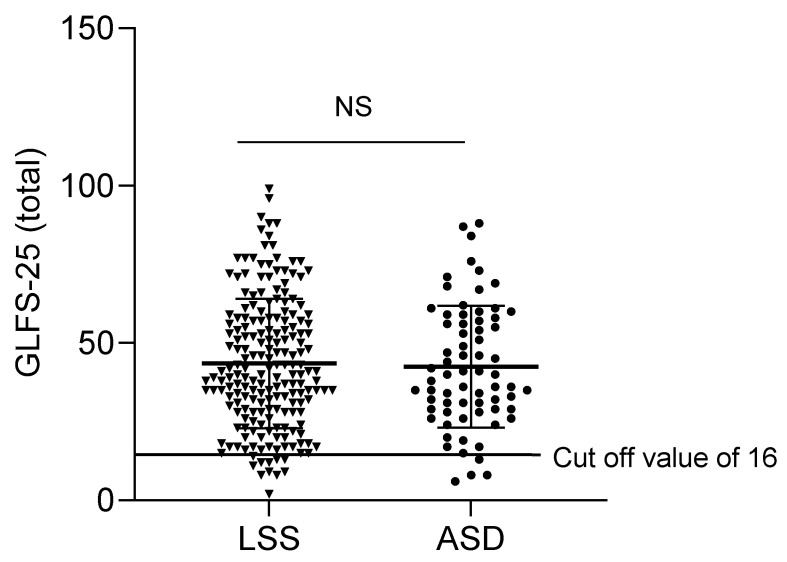
GLFS-25 total scores in patients with LSS or ASD. NS; No significant; LSS: Lumbar spinal stenosis; ASD: Adult spinal deformities; GLFS-25; Geriatric Locomotive Function Scale-25.

**Table 1 jcm-12-04345-t001:** Baseline characteristics and physical performance tests of patients with LSS and ASD.

Variable	LSS (N = 196)	ASD (N = 69)	*p*
Age (y)	74.6 ± 9.4	72.8 ± 7.3	0.13
Female/male gender (n)	172/24	62/7	0.83
BMI (kg/m^2^)	24.2 ± 2.1	23.8 ± 3.1	0.42
Test			
2-step test (points)	0.95 ± 0.4	1.04 ± 0.3	0.11
Stand-up test (points)	2.69 ± 1.5	2.86 ± 1.5	0.43

ASD, adult spinal deformity; LSS, Lumbar spinal stenosis; BMI, body mass index. The Kruskal–Wallis test was used to compare means between groups.

**Table 2 jcm-12-04345-t002:** Geriatric Locomotive Function Scale (GLFS-25) of patients with LSS or ASD.

Domain	Item	LSS	ASD	*p* *
Pain	1	0.96 ± 1.2	1.06 ± 1.1	0.51
	2	2.4 ± 1.0	2.4 ± 1.1	0.76
	3	2.3 ± 1.2	1.4 ± 1.3	<0.05
	4	2.2 ± 1.0	2.3 ± 0.9	0.63
Mobility	5	1.4 ± 1.1	1.4 ± 1.0	0.80
	6	1.2 ± 1.2	1.3 ± 1.2	0.73
	7	1.2 ± 1.2	1.4 ± 0.97	0.13
Self-care	**8**	**0.57 ± 0.86**	**0.84 ± 0.92**	**<0.05**
	9	1.3 ± 1.1	1.2 ± 1.1	0.43
	10	0.93 ± 1.1	0.79 ± 0.96	0.37
	**11**	**0.8 ± 0.96**	**1.2 ± 1.1**	**<0.05**
Mobility	12	2.1 ± 1.2	2.0 ± 1.2	0.69
	13	2.5 ± 1.2	2.4 ± 1.1	0.76
Self-care	**14**	**0.81 ± 1.0**	**1.2 ± 1.0**	**<0.05**
Mobility	15	2.7 ± 1.2	2.8 ± 1.2	0.63
Interpersonal interactions	16	1.6 ± 1.3	1.6 ± 1.2	0.76
Mobility	17	1.9 ± 1.4	2.0 ± 1.4	0.49
	18	1.7 ± 1.4	2.1 ± 1.3	0.06
Domestic life	19	1.5 ± 1.2	1.6 ± 1.9	0.35
	**20**	**2.1 ± 1.3**	**2.6 ± 1.1**	**<0.05**
Community	21	2.9 ± 1.1	3.0 ± 1.1	0.69
Interpersonal interactions	22	1.6 ± 1.4	1.5 ± 1.3	0.79
Community	23	2.2 ± 1.5	2.2 ± 1.4	0.91
Anxiety	24	1.3 ± 1.3	1.3 ± 1.1	0.69
	25	2.0 ± 1.3	2.0 ± 1.2	0.89

Bold letters and shaded rows denote items that were significantly worse in the ASD group compared to the LSS group. * *p* < 0.05 compared with ASD group.

**Table 3 jcm-12-04345-t003:** Spinopelvic parameters and correlation with score of Q20 in patients with ASD.

Variable	Preoperative	r-Value (Q20)	*p*-Value
PT (°)	37.4 ± 10.6		0.69
SS (°)	15.1 ± 13.4		0.33
LL (°)	9.1 ± 21.7		0.60
PI–LL (°)	42.1 ± 21.4	0.52 *	<0.05
SVA (mm)	124.8 ± 69.9		0.53
GT (°)	53.2 ± 17.5		0.85
TPA (°)	41.7 ± 14.9		0.90

GT, global tilt; LL, lumbar lordosis; PI, pelvic incidence; PT, pelvic tilt; SS, sacral slope; SVA, sagittal vertical axis; TPA, T1-pelvic angle. Interval and ratio values are presented as the mean ± standard deviation. * *p* < 0.05.

## Data Availability

We will be able to provide them if you contact corresponding author.

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
