# Peer review of "Comparative Study for Characteristics of Locomotive Syndrome in Patients with Lumbar Stenosis and Adult Spinal Deformity"

_jcm, 2023, doi:10.3390/jcm12134345_

Round 1

Reviewer 1 Report

In their manuscript, Authors investigate the prevalence and different characteristics of the “Locomotive syndrome” in patients affected by lumbar spine stenosis and adult spine deformities.

The manuscript is well written, and we found it of particular interest for clinical practice in that it underlines some aspects which could be taken in consideration during preoperative evaluation of patients harboring a degenerative spine disease. 

Authors confirmed that patients harboring lumbar spinal stenosis more often complain lower limb pain while it is interesting to note that patients affected by a degenerative deformity have more difficulty in the “dynamic” and load-bearing activity of daily living, like domestic and self-care activities. 

We think that the manuscript reaches the aim of the authors and so it is suitable. We only suggest the following modification:

-       It would be proper to add a “Limitation” section in which authors should emphasize that, being a retrospective observational study, there is a lack of postoperative data which could better highlight the qualitative variation of the GLFS-25 score after surgery. 

Author Response

Thank you for considering our manuscript, titled “Comparative study for Characteristics of Locomotive syndrome in patients with lumbar stenosis and adult spinal deformity”. We are grateful for the editorial and reviewers’ suggestions that have enabled revision and improvement of our manuscript.

We submit herewith a document with point-by-point responses to all the reviewers’ comments, along with an updated version of our manuscript, which has been revised in accordance with the comments. We hope that our responses are well received, and that the revised manuscript is now acceptable for publication in Journal of Clinical Medicine. Thank you for your time and thoughtful consideration of our work.

Reviewer1

In their manuscript, Authors investigate the prevalence and different characteristics of the “Locomotive syndrome” in patients affected by lumbar spine stenosis and adult spine deformities.

The manuscript is well written, and we found it of particular interest for clinical practice in that it underlines some aspects which could be taken in consideration during preoperative evaluation of patients harboring a degenerative spine disease. 

Authors confirmed that patients harboring lumbar spinal stenosis more often complain lower limb pain while it is interesting to note that patients affected by a degenerative deformity have more difficulty in the “dynamic” and load-bearing activity of daily living, like domestic and self-care activities. 

We think that the manuscript reaches the aim of the authors and so it is suitable. We only suggest the following modification:

 All authors appreciate the thorough review and favorable evaluations. We will continue the current study.

-       It would be proper to add a “Limitation” section in which authors should emphasize that, being a retrospective observational study, there is a lack of postoperative data which could better highlight the qualitative variation of the GLFS-25 score after surgery. 

→With respect to this suggestion, we have added limitations and further direction to research impact of surgery or conservative treatment on GLFS-25 scores and/or LS severity.

Reviewer 2 Report

 The paper is interesting & worthy of investigation 

1) abstract is well written 

2) introduction - the authors claim the study to plan the surgical plan .  The introduction should be rewritten . the study is only a comparative study between LSS & ASD .

3)material method - adequate 

4) discussion - good 

Author Response

Thank you for considering our manuscript, titled “Comparative study for Characteristics of Locomotive syndrome in patients with lumbar stenosis and adult spinal deformity”. We are grateful for the editorial and reviewers’ suggestions that have enabled revision and improvement of our manuscript.

We submit herewith a document with point-by-point responses to all the reviewers’ comments, along with an updated version of our manuscript, which has been revised in accordance with the comments. We hope that our responses are well received, and that the revised manuscript is now acceptable for publication in Journal of Clinical Medicine. Thank you for your time and thoughtful consideration of our work.

The paper is interesting & worthy of investigation

→ All authors appreciate the thorough review and favorable evaluations. We will continue the current study.

1) abstract is well written

2) introduction - the authors claim the study to plan the surgical plan .  The introduction should be rewritten . the study is only a comparative study between LSS & ASD .

→With respect to this suggestion, we have significantly rewritten Intro to include more about locomotion in ASD and LSS.

3)material method - adequate

4) discussion - good

Reviewer 3 Report

The authors describe a small correlation in the understanding of history and clinical presentation in patients with LSS and ASD which differs from patients without ASD. Finally, small detailed differences create awareness in the examiner and can help in a possible causal diagnosis. Whether the Q20 is correlated with the sagittal parameters is not clear even if a correlation was found. In addition, the reader will be interested to know why, instead of the total score of this screening instrument, he has to pay attention to the Q20? Here it would be helpful a clinical derivation of the result, should be asked specifically to the differentiation, or is such a result lost if only the questionnaire is filled out. What can we as clinicians take away from this? However, limitations of the study are missing, only through these can an experienced reader become aware of which bias he has to pay attention to.Nevertheless, it should be discussed whether a therapeutic consequence should be derived for such a patient population. In my opinion, however, it is a reasonable question and investigation to find out when the deterioration in this score starts and when an early intervention would be indicated. Say where can we collect data within the framework of big data to effect and possibly counteract early intervention.

Needs minor revisions

Author Response

Thank you for considering our manuscript, titled “Comparative study for Characteristics of Locomotive syndrome in patients with lumbar stenosis and adult spinal deformity”. We are grateful for the editorial and reviewers’ suggestions that have enabled revision and improvement of our manuscript.

We submit herewith a document with point-by-point responses to all the reviewers’ comments, along with an updated version of our manuscript, which has been revised in accordance with the comments. We hope that our responses are well received, and that the revised manuscript is now acceptable for publication in Journal of Clinical Medicine. Thank you for your time and thoughtful consideration of our work.

Reviewre2

The authors describe a small correlation in the understanding of history and clinical presentation in patients with LSS and ASD which differs from patients without ASD. Finally, small detailed differences create awareness in the examiner and can help in a possible causal diagnosis. Whether the Q20 is correlated with the sagittal parameters is not clear even if a correlation was found. In addition, the reader will be interested to know why, instead of the total score of this screening instrument, he has to pay attention to the Q20? Here it would be helpful a clinical derivation of the result, should be asked specifically to the differentiation, or is such a result lost if only the questionnaire is filled out. What can we as clinicians take away from this?

→ Sorry for the confusion due to the inadequate description of the results. We examined the correlation between spinopelvic parameters and scores on all questions, including total score, and the only significant correlation in this study was with Q20. Now, we have added this information in result section. Additionally, we have added the discussion why we did not find a correlation between any questionnaire other than Q20 and the spinal pelvic parameters in this study.

 However, limitations of the study are missing, only through these can an experienced reader become aware of which bias he has to pay attention to.Nevertheless, it should be discussed whether a therapeutic consequence should be derived for such a patient population. In my opinion, however, it is a reasonable question and investigation to find out when the deterioration in this score starts and when an early intervention would be indicated. Say where can we collect data within the framework of big data to effect and possibly counteract early intervention.

→We completely agree early diagnosis and early intervention are important for the treatment of LS. Now, we have added limitations of the study and discussions in manuscript.